

# A new merged dataset for analyzing clouds, precipitation and atmospheric parameters based on ERA5 reanalysis data and the measurements of TRMM PR and VIRS

Lilu Sun[1], Yunfei Fu[1]

[1]School of Earth and Space Sciences, University of Science and Technology of China, Hefei, 230026, China

*Correspondence to: Yunfei Fu (fyf@ustc.edu.cn)*

**Abstract.** Clouds and precipitation have vital roles in the global hydrological cycle and the radiation budget of the atmosphere–Earth system and are closely related to both the regional and global climate. Changes in the status of the atmosphere inside clouds and precipitation systems are also important, but the use of multi-source datasets is hampered by their different spatial and temporal resolutions. We merged the precipitation parameters measured by the Tropical Rainfall Measuring Mission (TRMM) Precipitation Radar (PR) with the multi-channel cloud-top radiance measured by the Visible and Infrared Scanner (VIRS) and atmospheric parameters in the ERA5 reanalysis dataset. The merging of pixels between the precipitation parameters and multi-channel cloud-top radiance was shown to be reasonable. The 1B01-2A25 dataset of pixel-merged data (1B01-2A25-PMD) contains cloud parameters for each PR pixel. The 1B01-2A25 gridded dataset (1B01-2A25-GD) was merged spatially with the ERA5 reanalysis data. The statistical results indicate that gridding has no unacceptable influence on the parameters in the 1B01-2A25-PMD. In one orbit, the difference in the mean value of the near-surface rain rate and the signals measured by the VIRS was no more than 0.87 and the standard deviation was no more than 2.38. The 1B01-2A25-GD and ERA5 datasets were spatiotemporally collocated to establish the merged 1B01-2A25 gridded dataset (M-1B01-2A25-GD). Three case studies of typical cloud and precipitation events were analyzed to illustrate the practical use of the M-1B01-2A25-GD. This new merged gridded dataset can be used to study clouds and precipitation systems and provides a perfect opportunity for multi-source data analysis and model simulations. The data which were used in this paper are freely available at http://doi.org/10.5281/zenodo.4458868 (Sun and Fu, 2021).

## 1 Introduction



Clouds and precipitation are the result of interactions among several different atmospheric parameters.
They have a crucial role in global hydrological and energy cycles which can be used to represent changes
in the Earth's weather and climate (Hartmann and Short, 1980; Liou, 1986; Wetherald and Manabe, 1988;
Baker, 1997; Houze, 1997; Roscow et al., 1999; Oki and Kanae, 2006; Lau and Wu, 2010; Fu and Zhang,
2018). More than two-thirds of all precipitation falls in the tropics and subtropics. The atmosphere
obtains energy from the release of the latent heat of condensation during precipitation and this is one of
the main drivers of atmospheric circulation at low latitudes. Precipitating clouds also influence the
Earth's radiation budget (Fu et al., 2006; Sassen et al., 2009; Kienast-Sjögren et al., 2016; Fu and Zhang,
2018; Gao et al., 2018). Atmospheric parameters (e.g., the temperature, pressure and wind fields) can be
used to demonstrate changes in atmospheric status and the development of cloud and precipitation
systems (Wang et al., 2015; Zheng et al., 2015; Pan and Fu, 2015; Xia and Fu, 2016; Wang et al., 2017).
Analysis of the complex distribution of clouds and precipitation, especially their 3D structures, helps to
require a detailed understanding of the microphysical processes and thermodynamic structure of clouds.
The horizontal structure represents the extent of the system, whereas the thermodynamic structure and
microphysical processes inside clouds during the phase transition of water can be represented by the
vertical structure (Houze, 1981; Szoke et al., 1986; Hobbs, 1991; Zipser and Lutz, 1994; Yuter and Houze,
1995; Liu and Fu, 2001; Luo et al., 2009). A comprehensive understanding of the 3D structure of
precipitation and clouds will improve precipitation retrieval algorithms and model simulations (Wilheit
et al., 1977; Petty, 1994; Kummerow et al., 1996; Olson et al., 1996; Iguchi et al., 2000; Tustison et al.,
2002; Min et al., 2013).
Satellite remote sensing technology has developed rapidly in recent years. The first space-borne active
radar is the Precipitation Radar (PR) onboard the Tropical Rainfall Measurement Mission (TRMM)
(Kummerow et al., 1998; Kummerow et al., 2000; Kozu et al., 2001). The main aim of the TRMM is the
effective observation of precipitation and energy exchange in tropical and subtropical regions. The
unique instruments onboard the satellite provide an excellent opportunity to study the 3D structure of
precipitation (Nesbitt et al., 1999; Schumacher and Houze, 2003; Durden et al., 2003; Li and Fu, 2005;
Liu and Zipser, 2009), whereas the Visible and Infrared Scanner (VIRS) provides characteristics of cloud
parameters near the cloud-top (Liu and Fu, 2010; Fu, 2014; Yang et al., 2014; Chen and Fu, 2017).
Many studies of the 3D structure of clouds and precipitation have been based on TRMM measurements



(Fu and Liu, 2001; Lu et al., 2016; Wang and Fu, 2017; Luo et al., 2020). Chen and Fu (2015) focused
on the characteristics of precipitation and thermal infrared signals of clouds in typhoon and non-typhoon
precipitation in eastern Asia by analyzing merged data between the PR and the VIRS pixels. Results
showed that the most intense typhoon precipitation occured on the ocean surface in eastern Asia. Non-
typhoon precipitation was the main type of precipitation in the rainy season, but typhoon precipitation
made the largest contributed to the total amount of precipitation. Yang et al. (2014) statistically analyzed
10-year data from the TRMM PR and VIRS and found that the cloud height and thickness of anvil clouds
over land were always greater than those over the sea surface, and anvil clouds had a greater optical
thickness over the land surface and more small ice particles.
Liu and Fu (2001) classified tropical precipitation into two main types (deep convective and stratiform)
using the main component method. The mean rain rate profiles of deep convective rain had four layers,
whereas stratiform precipitation had only three. Differences in the slopes of the mean profiles can be
used to illustrate the microphysical processes at different vertical structure of the rainfall. Liu and Fu
(2007) compared rain rate profiles from the Tibetan Plateau, eastern Asia and the tropics. It was found
that there was little stratiform precipitation on the Tibetan Plateau, but deep weak convective
precipitation was frequently detected in this region. The slope of the mean deep weak convective profile
was greater than that for convective precipitation, which meant that more latent heat was released to heat
the middle and upper atmosphere. Fu et al. (2008) found that the tops of precipitating clouds were about
4–6 km higher than that in the surrounding area. The difference in the clouds tops height led to a tower-
like structure over the Tibetan Plateau, which can heat the upper troposphere more easily.
Recent researchs has investigated the status of the atmosphere in clouds and precipitation systems (Wang
and Fu, 2017; Li et al., 2018). In addition to observations from surface meteorological stations,
atmospheric parameters can be obtained from sounding balloons, sensors onboard planes and model
simulations. Because of the uneven distribution of ground meteorological stations, the unpredictable
routes of sounding balloons and the limited detection area of planes. We therefore analyzed the
characteristics and distribution of atmospheric parameters inside clouds and precipitation systems using
the ERA5 reanalysis dataset.
It is impossible to obtain simultaneous and comprehensive information using a single detection method
or dataset as a result of the different ways of obtaining the cloud, precipitation and atmospheric



parameters. The optimum choice to overcome this problem is to combine multiple remote sensing
measurements and multi-source datasets (Fu et al., 2013; Chen and Fu, 2017). Hawkins et al. (2008)
combined multiple source datasets, including satellite cloud imagery captured by cloud profile radar and
stationary satellites, observations from experimental planes and model simulations. The merged data can
be used to determine the vertical structure of precipitating clouds and the atmospheric parameters in
typhoon, frontal and thermal convective precipitation systems.
VIRS pixels can be merged onto PR pixels in a similar way to verify precipitating clouds. The results of
this type of merger have shown that the reflectivity in the visible channel near the precipitating cloud-
top is bigger than 0.5 in the intertropical convergence zone, convergence zone of the southern hemisphere,
the monsoon region of Asia, the tropical regions of Africa, North America and South America. The
reflectivity on land is greater than that on the sea surface (Fu et al., 2011). Liu and Fu (2010) verified
precipitating clouds from several typhoon and frontal precipitation events in eastern China based on the
same merged dataset. Precipitating clouds cannot be classified precisely by relying on only the thermal
infrared brightness temperature and other parameters are needed to improve the accuracy, such as the
ratio between the signals in the visible and near infrared channels.
The various temporal and spatial resolutions of different datasets can cause problems and it is better to
merge data from multiple instruments onboard the same satellite, such as the TRMM. The ERA5
reanalysis dataset has a suitable temporospatial resolution for merging with the TRMM data to
supplement the atmospheric parameters. This new merged dataset includes comprehensive parameters
that can be used to analyze the features of the precipitation and clouds systems.
We merged TRMM PR and VIRS measurements with the ERA5 reanalysis data at the same
spatiotemporal resolution to establish a new dataset of precipitation, cloud and atmospheric parameters.
Section 2 describes the data and merging methods. Section 3 presents the main results about the influence
of the merger on the original data during the merging process and the pratical applications of the new
merged dataset. Section 4 discusses the advantages of the dataset and future work in the progress. Access
to the dataset is introduced in Section 5 and conclusions are presented in Section 6.



## 2    Data and methods

### 2.1    Tropical Rainfall Measurement Mission

The TRMM was launched in November 1997 as a joint mission between the US National Aeronautics and Space Administration (NASA) and the Japan Aerospace Exploration Agency (JAXA). The objectives of the TRMM are to obtain satellite measuremenets of rainfall and energy exchange in tropical and subtropical regions (https://trmm.gsfc.nasa.gov/overview_dir/background.html). The TRMM is a non-solar synchronous polar orbiting satellite in a 350 km (402 km after an orbital boost on 7 August 2001) circular orbit with an inclination angle of 35°. The TRMM observes a specific location between 38° S and 38° N every 45 days. One complete scan of the orbit takes about 96 minutes, so there are 16 orbits in one day (Simpson et al., 1996; Kummerow et al., 1998).

### 2.2    PR and 2A25 dataset

The PR was the first space-borne precipitation radar onboard the TRMM. It is a 128-element active phased array system operating at 13.8 GHz (Kozu et al., 2001). The PR antenna scans in the cross-track direction over 17° in a 215 km (220 km after the orbital boost) swath width. There are 49 pixels in each scan line. The PR measures the spatial distribution of the intensity of precipitation from mean sea-level to 20 km (80 layers in total) and has a horizontal resolution of 4.5 km at nadir (5 km after the orbital boost). The vertical resolution of the PR is 0.25 km.

The 2A25 data produced by NASA Goddard Space Flight Center is the second-level data product of the TRMM PR. The dataset includes the scan time, geolocation information, rain type, 3D rain rate and so on. The TRMM PR algorithm classifies the type of rain into convective, stratiform and "other" (Awaka et al., 1997; Hayasaka et al., 1998).

### 2.3    VIRS and 1B01 datasets

The VIRS antenna scans in the cross-track direction over 45° in a 720 km (833 km after the orbital boost) swath width. There are 261 pixels in each scan line and the horizontal resolution is 2.2 km at nadir (2.4 km after the orbital boost). The VIRS receives upward radiation at five wavelengths ranging from the visible to the far infrared: 0.63 µm (CH1), 1.6 µm (CH2), 3.7 µm (CH3), 10.8 µm (CH4) and 12.0 µm (CH5).

The 1B01 dataset is the first-level data product of the VIRS. The 1B01 data include the reflectivity in



CH1 and CH2 (RF1 and RF2) and the infrared radiation brightness temperature ($TB_{3.7}$, $TB_{10.8}$ and $TB_{12.0}$)
in CH3, CH4 and CH5, which are calibrated from the spectral signals measures by the VIRS.

**2.4   ERA5 reanalysis dataset**

The ERA5 renalysis dataset is the fifth (latest) generation of global atmospheric reanalysis datasets
produced by the European Centre for Medium-Range Weather Forecasts. The ERA5 dataset is based on
the Integrated Forecasting System Cy41r2 model and assimilates more model simulation outputs and
observational results. The ERA5 dataset is superior to previous versions in terms of its hourly output,
finer spatial resolution (0.25°) and abundant parameters (Zhao et al., 2019; Hersbach et al., 2020). We
used the hourly atmospheric parameters (divergence, geopotential height, specific humidity, wind field,
vertical velocity and temperature) on pressure levels for 32 layers from 1000 to 10 hPa. All the pressure
layers are used during the data merging, except the uppermost pressure layers from 1 to 7 hPa, which are
rarely used in studies.

**2.5   2A25 and 1B01 merged data**

The characteristics of precipitation and clouds are shown by identifying precipitating clouds from the
PR and detecting radiance near the top of the cloud by the VIRS (Liu and Fu, 2010; Fu et al., 2011;
Chen et al., 2018). The 2A25 and 1B01 data products (derived from the TRMM PR and VIRS,
respectively) can be collocated to establish a merged dataset to provide comprehensive information
about precipitation and clouds systems. The feasibility of data merging depends largely on the sensor
settings, such as the temporal sampling rate, the synchronism of detection and the spatial resolution.
The PR and the VIRS are both the main sensors onboard the TRMM. Despite the difference in spatial
resolution between 1B01 and 2A25, the time lag between detections of the same target is less than 1
minute. The similar cross-track scanning modes make it reasonable to consider that the PR and VIRS
are roughly synchronous in their detection area. It is therefore feasible to combine these two orbit-level
data products. Spatial merging is the only process that needs to be taken into account because of the
quasi-synchronicity between the TRMM PR and VIRS.
On account of the diverse orbital swath widths and spatial resolutions, the horizontal resolution of the
1B01 is decreased to match that of the 2A25 so that data merging can be easily achieved. The VIRS
pixels are merged onto the PR pixels through a weight-averaged method. The spectral signals are





calculated near the PR pixel and there are usually about seven VIRS pixels near one PR pixel (Fu et al.,
2011). The primary data of the 1B01 include the reflectivity at CH1 and CH2 (RF1 and RF2), the
equivalent brightness temperature of a black body at CH3, CH4 and CH5 ($TB_{3.7}$, $TB_{10.8}$ and $TB_{12.0}$), the
instantaneous near-surface rain rate and the vertical structure of precipitation in 2A25. All the VIRS
signals are within the resolution of the PR pixel which can be used to study the characteristics of
precipitating clouds. The 1B01-2A25 pixel-merged data (1B01-2A25-PMD) are then established (Liu
and Fu, 2010; Chen and Fu, 2015; Chen and Fu, 2017).
**2.6  Gridding of the 1B01-2A25-PMD**
Because the 1B01-2A25-PMD contains orbit-level data, data gridding is necessary to merge these data
with the ERA5 data. The 1B01-2A25-PMD is gridded between 40° S and 40° N and the spatial
resolution is 0.25°, in agreement with the resolution of the ERA5 data. Taking the near-surface rain rate
as an example, we first sum the near-surface rain rate of the pixels in the same grid and then counting
the number of precipitating pixels in one grid and dividing the total rain rate by the total number of
precipitating pixels to obtain the near-surface rain rate of one grid. The grid-level dataset, namely, the
1B01-2A25 gridded data (1B01-2A25-GD) is calculated in the same way. The dataset includes the
gridded spectral signals measured by the VIRS in each PR pixel. The precipitation is classified into
three types: total precipitation, convective precipitation, and stratiform precipitation. The PR is a Ku-
band radar with a working frequency of 13.8 GHz and a wavelength of 2.2 cm. The sensitivity of the
PR is 16 dBZ and the minimum rain rate it can detect is about 0.4 mm h$^{-1}$. Only pixels with a near-
surface rain rate greater than 0.4 mm h$^{-1}$ are calculated.
**2.7  Merging the ERA5 and 1B01-2A25-GD**
We merged the ERA5 data onto the 1B01-2A25-GD grid to explore the atmospheric status of the
precipitation and clouds systems measured by the sensors onboard the TRMM. The ERA5 dataset has
hourly outputs and the ERA5 parameters were therefore interpolated according to the grid times in the
1B01-2A25-GD. The geolocation of the ERA5 grid was selected to match that of the 1B01-2A25-GD.
The merged 1B01-2A25 gridded dataset (M-1B01-2A25-GD) was therefore obtained, which includes
geolocation information, the time of the grid and parameters for precipitation, clouds and the
atmosphere.



## 3 Results

### 3.1 Evaluation of the 1B01-2A25-PMD

The horizontal resolution of the 1B01 is decreased in the 1B01-2A25-PMD as a result of merging onto

the PR pixels. The detection field of the merged data is narrower than that in the 1B01 and is about the

same as the swath width of the PR. The influence on the original data in 1B01 after merging is

presented based on comparisons between the probability distribution functions (PDFs). Taking an

arbitrary orbit for example, orbit 37362 on 5 June 2004. Comparisons are made between the signals in

five spectral channels in the 1B01 and 1B01-2A25-PMD. Figure 1a–e represent the PDFs of RF1, RF2,

$TB_{3.7}$, $TB_{10.8}$ and $TB_{12.0}$ in the 1B01 and 1B01-2A25-PMD, respectively.

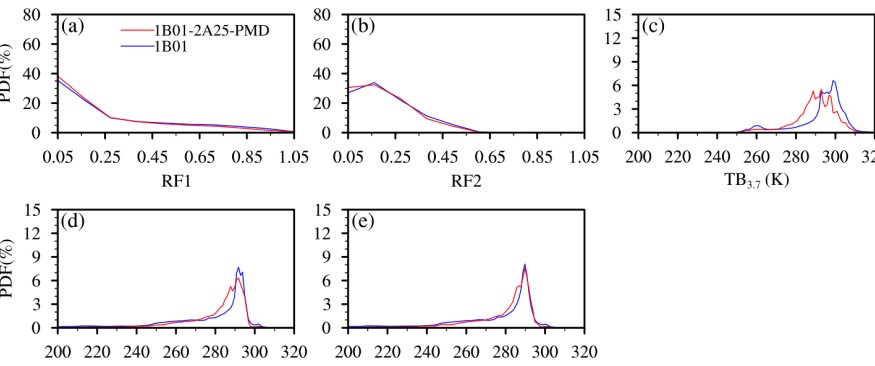

**Figure 1: PDFs of (a) RF1, (b) RF2, (c) $TB_{3.7}$, (d) $TB_{10.8}$ and (e) $TB_{12.0}$ in the 1B01 (blue line) and 1B01-2A25-PMD (red line) in orbit number 37362 on 5 June 2004.**

Figure 1a and 1b show that there is no clear change in RF1 and RF2 after merging. The shape of the

PDFs are nearly the same in the 1B01-2A25-PMD and 1B01, with the reflectivity ranging from 0.05 to

1.05, but concentrate in the range 0.05–0.3. By contrast, there is a clear variation in $TB_{3.7}$ after merging

(Fig. 1c). The PDF of the 1B01 is a single peak with a maximum at 300 K and a sub-peak at 290 K. The

PDF of the 1B01-2A25-PMD shows a multi-peak shape, with the peaks mostly in range 280–300 K.

$TB_{3.7}$ ranges from 250 to 320 K in both the 1B01 and 1B01-2A25-PMD. The difference in $TB_{3.7}$ between

the 1B01 and 1B01-2A25-PMD is probably caused by the uneven cloud distribution. $TB_{10.8}$ and $TB_{12.0}$

have the same distribution after merging. The signals vary from 240 to 300 K and the maxima of the

PDFs are both at about 290 K. Although the PDFs of the original and merged datasets are similar, there

are slight differences in the peak values. The PDFs of the 1B01-2A25-PMD show lower peaks as a result

of data averaging inside the grid (Fig. 1c and 1d).
Two regions are selected to analyze the distinctive changes in $TB_{3.7}$ after merging onto the PR pixels: (1)
a region that mainly contains cloudy pixels (the cloudy region), and (2) a region that mainly contains
clear sky pixels (the clear sky region). When the radiance brightness temperature is greater than 290 K,
the pixel is classified as a clear sky pixel, whereas cloudy pixels are classified as those pixels in which
the near-surface rain rate is greater than 0.4 mm h$^{-1}$.

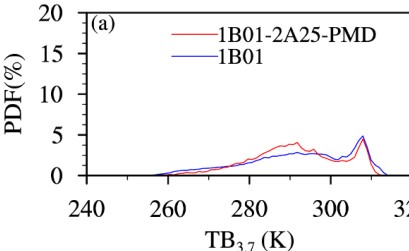 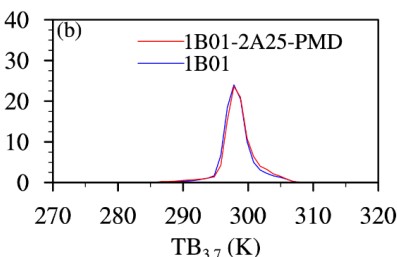


**Figure 2: PDFs of $TB_{3.7}$ in (a) cloudy regions and (b) clear sky regions in the 1B01 (blue line) and 1B01-2A25-**
**PMD (red line) in orbit number 37362 on 5 June 2004.**
Fig. 2 shows the PDFs of $TB_{3.7}$ in these two regions. $TB_{3.7}$ ranges from 260 to 315 K in the cloudy region
and the maximum value (310 K) is almost the same in both the 1B01 and 1B01-2A25-PMD datasets (Fig.
2a). This similarity is also seen in the PDF of $TB_{3.7}$ in the clear sky region, where $TB_{3.7}$ varies between
290 and 310 K with a single-peak structure and a maximum at 300 K (Fig. 2b). In a word, the merging
process between the PR and the VIRS pixels is considered lead to no dramatic variations on the original
data.
**3.2   Evaluation of the 1B01-2A25-GD**
The 1B01-2A25-PMD was gridded to match the ERA5 reanalysis data spatially and this dataset is
referred to as the 1B01-2A25-Gridded Data (1B01-2A25-GD). Data processing studies have shown that
the precision of the spatial resolution of a dataset can affect the accuracy and physical characteristics of
the data (Heng and Fu, 2014) and therefore it is essential to evaluate the effects of the gridding process.
As an example, Figure 3 compares the near-surface rain rate and the CH1–CH5 signals of the original
and gridded data for orbit 37362 on 5 June 2004 in both the 1B01-2A25-PMD and 1B01-2A25-GD.



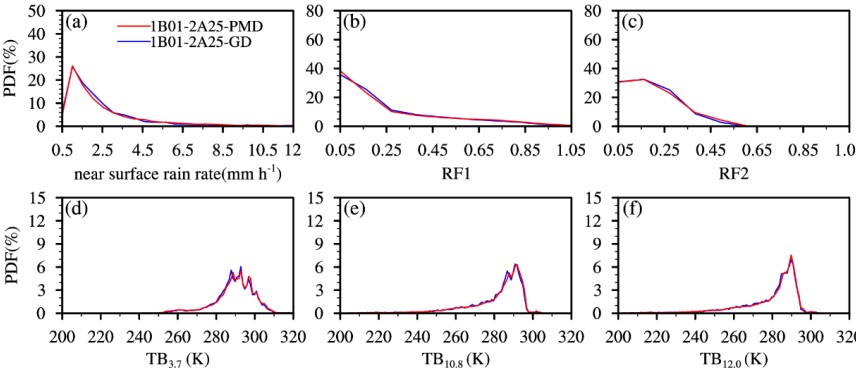


**Figure 3: PDFs of (a) the near-surface rain rate, (b) RF1, (c) RF2, (d) TB$_{3.7}$, (e) TB$_{10.8}$ and (f) TB$_{12.0}$ in the 1B01-2A25-PMD (red line) and 1B01-2A25-GD (blue line) in orbit number 37362 on 5 June 2004.**

Figure 3a shows that the PDF of the near-surface rain rate in the 1B01-2A25-PMD has a single peak and
the rain rate mainly ranges from 1 to 3 mm h$^{-1}$. The PDF of the gridded data is nearly the same as that
of the original data, so gridding has little influence on the near-surface rain rate. The PDFs for RF1 and
RF2 are also largely unchanged. RF1 varies from 0.05 to 1.05 and RF2 is mainly in the range 0.05–0.55
(Fig. 3b, 3c). The brightness temperature (TB$_{3.7}$, TB$_{10.8}$ and TB$_{12.0}$) ranges from 240 to 310 K and there
is little difference in the PDFs of the 1B01-2A25-PMD and 1B01-2A25-GD. The PDF of TB$_{3.7}$ has a
multi-peak structure, whereas the PDFs of TB$_{10.8}$ and TB$_{12.0}$ are both single peaks (Fig. 3d–3f). Gridding
therefore does not result in dramatic variations of the parameters in the 1B01-2A25-PMD.
Statistical calculations were carried out to quantify the influence of gridding on the 1B01-2A25-PMD.
Table 1 shows the mean, standard deviation (STD) and the corresponding differences of the near-surface
rain rate and the signals from the five channels in the 1B01-2A25-PMD and 1B01-2A25-GD for orbit
37362 on 5 June 2004.
**Table 1. Comparisons of statistical mean and standard deviation (STD) of the near-surface rain rate and**
**signals from five channels in the 1B01-2A25-PMD and 1B01-2A25-GD for orbit number 37362 on 5 June 2004.**

|  | Before gridding | | After gridding | | Difference | |
|---|---|---|---|---|---|---|
|  | Mean | STD | Mean | STD | Mean | STD |
| RF1 | 0.112 | 0.195 | 0.110 | 0.189 | 0.002 | 0.006 |
| RF2 | 0.083 | 0.126 | 0.082 | 0.123 | 0.001 | 0.003 |





| | | | | | | |
| --- | --- | --- | --- | --- | --- | --- |
| $TB_{3.7}$ | 288.805 | 11.928 | 288.682 | 10.631 | 0.123 | 1.297 |
| $TB_{10.8}$ | 280.516 | 16.120 | 280.521 | 15.397 | -0.005 | 0.723 |
| $TB_{12.0}$ | 278.584 | 16.574 | 278.628 | 15.882 | -0.044 | 0.692 |
| Near-surface rain rate | 3.072 | 5.121 | 2.209 | 2.745 | 0.861 | 2.376 |

These results show that the difference in the mean value in this orbit is no more than 0.87 and the STD
is no more than 2.38. The differences in the near-surface rain rate are always larger than the differences
in the signals for the five channels, although all the differences are acceptable. The statistical results
quantitively verifiy that the slight influence caused by gridding can be neglected and the parameters in
the 1B01-2A25-GD are reliable.
**3.3  Evaluation of the M-1B01-2A25-GD**
The ERA5 atmospheric parameters are merged with the 1B01-2A25-GD to establish a new gridded
dataset (M-1B01-2A25-GD). The merging process among the 1B01, 2A25 and ERA5 datasets is based
on the 2A25, so the detection field is about the same as that measured by the TRMM PR in the new
merged dataset. Taking orbit 37362 on 5 June 2004 in the M-1B01-2A25-GD as an example.

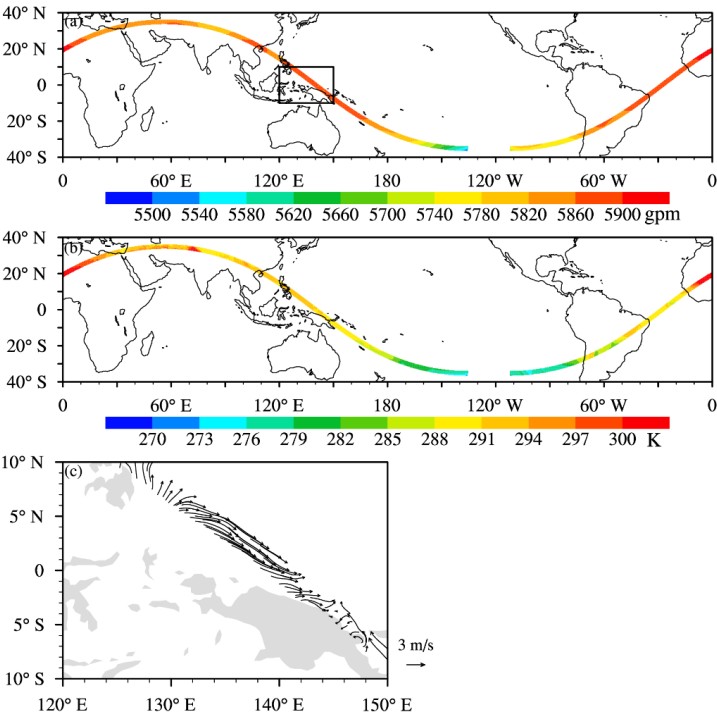

**Figure 4: Distribution of (a) the geopotential height at 500 hPa, (b) the temperature at 850 hPa and (c) the wind field at 1000 hPa in orbit number 37362 on 5 June 2004 in the M-1B01-2A25-GD.**

Figure 4 shows the geopotential height at 500 hPa, the temperature at 850 hPa and the wind field at 1000 hPa. The geopotential height is mainly in the range 5540–5900 gpm and the temperature ranges from 270 to 300 K (Fig. 4a, 4b). The box in Fig. 4a shows an enlarged view of the wind field. A strong northwesterly wind appears on the sea surface. The parameters for precipitation, clouds and the atmospheric status provided by the M-1B01-2A25-GD can therefore be used to study the properties of precipitation and clouds systems.

## 3.4 Applications of the M-1B01-2A25-GD

Evaluations of the M-1B01-2A25-GD show that the merging process does not seriously distort the original data. The new dataset can therefore be used to analyze different types of precipitation and clouds systems. To illustrate the applicability of this new dataset to a variety of different scenarios, we selected precipitation on the trumpet-shaped topography of the Tibetan Plateau, typhoon Rananim precipitation and frontal precipitation in eastern China for further analysis.

### 3.4.1 Precipitation on the trumpet-shaped topography of the southern Tibetan Plateau

The trumpet-shaped topography of the southern Tibetan Plateau (90°–100° E, 22°–32° N) is the one of

the main channel for the transport of water vapor in this region. The topography here is complex and

there is a clear difference in altitude in the trumpet-shaped area from the hills below the Tibetan

Plateau to the steep slopes and the central Tibetan Plateau. Define these three types of topography as

the foreland under the Tibetan Plateau (FTP), the slope of the trumpet-shaped area (STS) and the

central part of the Tibetan Plateau (CTP).

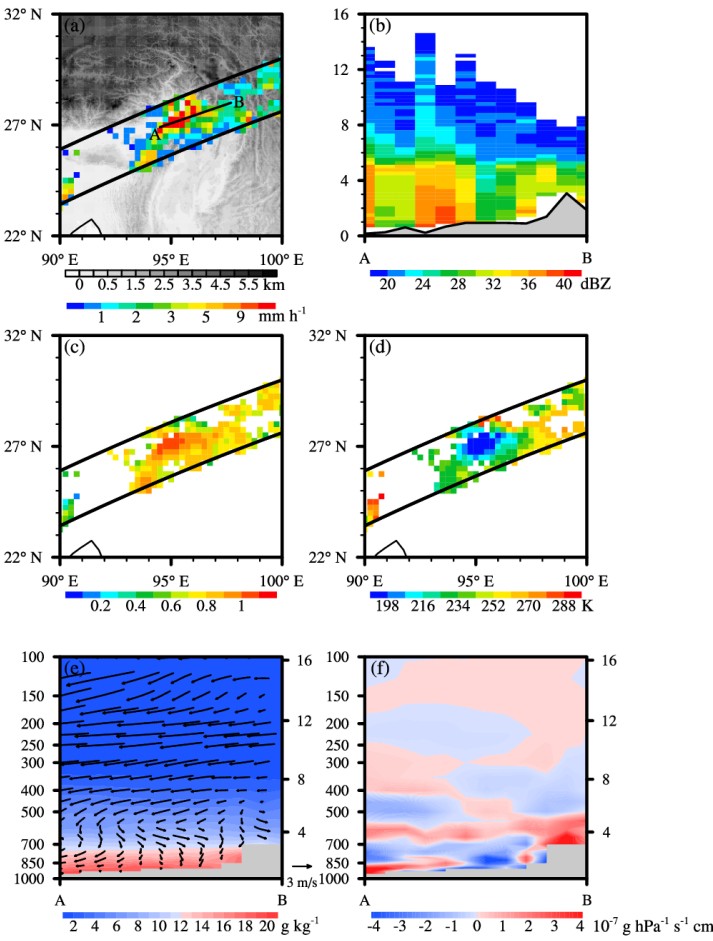

**Figure 5: Case study of precipitation in the trumpet-shaped topography on the Tibetan Plateau on 2 July 2004 (orbit number 37786). (a) The near-surface rain rate, (b) vertical cross-sections of the precipitation reflectivity factor, (c) RF1, (d) TB$_{10.8}$, vertical cross-sections of (e) the wind vectors and specific humidity and (f) the divergence of water vapor flux along line A-B in panel (a).**



The precipitation occurred on 2 July 2004 (orbit number 37786). Intense rainfall mainly occurs on the
STS with a maximum precipitation intensity over 13 mm h$^{-1}$. Fig. 5a shows the near-surface rain rate in
the rain belt and Fig. 5b shows the vertical cross-section of the precipitation reflectivity factor in the
direction of the heavy rainfall center shown in Fig. 5a (line A–B). The altitude varies by 2 km from A to
B. The storm top height (STH) refers to the height at which the precipitation reflectivity factor of the
three continuous layers is greater than 16 dBZ in the precipitation profiles, decreases gradually and the
strong reflectivity factor weakens over the CTP. The intensity of precipitation on the STS from the
atmosphere to the land surface first increases and then decreases. There is usually a strong precipitation
reflectivity factor 3 km above the land surface. However, the intensity of precipitation on CTP is weaker
than that in the FTP. Fu et al. (2017) showed that the elevation of water vapor by the complex topography
leads to precipitation on the STS. However, the intensity of precipitation is weak because the water vapor
column on the CTP is inadequate due to earlier precipitation on the STS.
The physical characteristics of the precipitating clouds are different as a result of the differences in the
intensity of precipitation. Fig. 5c shows RF1, which ranges between 0.5 and 1.0, with high values in the
areas where precipitation is heavy. A high value of RF1 means there are mainly small particles near the
top of the precipitation clouds. $TB_{10.8}$ varies from 190 to 210 K in the region of intense precipitation,
which shows that the cloud-top is high. The cloud-top is slightly lower around the precipitating cloud
where $TB_{10.8}$ is high (Fig. 5d). In addition to the topographic elevation of water vapor, it is also necessary
to understand the status of the atmosphere, including the wind field and water vapor conditions.
There is a wet band below 700 hPa with a specific humidity over 15 g kg$^{-1}$. The specific humidity
decreases in the lower layers of the atmosphere from the FTP to the STS and extending to the CTP. A
northeasterly airflow prevails below 700 hPa from the CTP via the STS to the FTP. Between 700 and
500 hPa, this northeasterly airflow turns to the northwest over the CTP. Above 500 hPa, the airflow is
mainly westerly with a higher speed. The water vapor content becomes less favorable for precipitation
in the higher atmospheric layers (Fig. 5e). The better water vapor conditions in the lower atmosphere are
important for the precipitation process. The upward airflow toward the CTP brings abundant water vapor
from the FTP via the STS to the CTP.
The water vapor flux divergence (WVFD) is an important parameter used to describe the status of water
vapor transportation in the atmosphere. The WVFD of the FTP is about $1\text{-}3\times10^{-7}$ g hPa$^{-1}$ s$^{-1}$ cm$^{-2}$, which



means that the water vapor is diverging. The divergence belt extends to the front of the STS. The WVFD
changes to positive on the STS (from $-2 \times 10^{-7}$ to 0 g hPa$^{-1}$ s$^{-1}$ cm$^{-2}$), so the water vapor here is in a
convergence field. There is strong divergence on the CTP with a maximum more than $4 \times 10^{-7}$ g hPa$^{-1}$ s$^{-1}$
cm$^{-2}$. Fig. 5f shows the complex water vapor transportation processes from the FTP to the CTP. Most of
the transport and exchange of water vapor occurs in the lower atmosphere. There is more water vapor in
the FTP and STS than in the CTP and the intensity of precipitation is heavy in the STS. The elevated
topography increases the transportation of water vapor and chance of precipitation, the wind speed and
convergence of water vapor indeed favor heavy rainfall over the STS.
**3.4.2 Typhoon Rananim precipitation**
Second precipitation example, Typhoon Rananim, occurred in the western Pacific (16°–26° N, 122°–132°
E) at orbit 38395 on 10 August 2004.

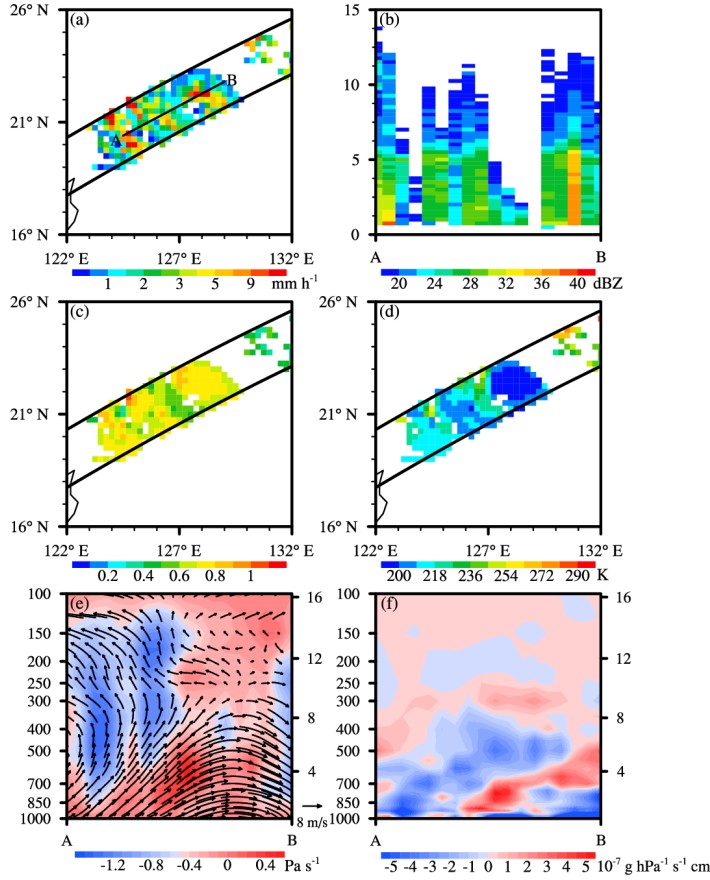




**Figure 6: Case study of Typhoon Rananim precipitation on 10 August 2004 (orbit number 38395). (a) The**
**near-surface rain rate, (b)vertical cross-sections of the precipitation reflectivity factor, (c) RF1, (d) TB$_{10.8}$,**
**vertical cross-sections of (e) the wind vectors and vertical velocity and (f) the divergence of water vapor flux**
**along line A-B in panel (a).**
The eye of a typhoon is an important indicator over the whole lifetime of the typhoon. The eye of the
typhoon shown in Fig. 6a means the typhoon is on the mature stage and the cross-section along line A–
B crosses the middle of the eye. The STH is higher than 13 km in the eye wall region, which means that
strong convection occurs here, but is lower than 5 km in the eye of the typhoon. The maximum
precipitation reflectivity factor exceeds 35 dBZ, but the precipitation intensity inside the eye is weak,
resulting in a low precipitation reflectivity factor and STH (Fig. 6b).
RF1 varies from 0.5 to 1. The particles near the precipitating cloud-top are small near the eye wall, which
leads to strong scatter in the visible spectrum (Fig. 6c). The distribution of TB$_{10.8}$ shows the height of the
precipitating cloud-top, which also indirectly represents convection inside the precipitation system.
TB$_{10.8}$ is less than 235 K around the eye of the typhoon. A low brightness temperature means a high
precipitating cloud-top. The STH is low and the cloud-top is high in some regions as a result of cirrus
clouds (Fig. 6d). The maximum descending speed in the typhoon eye is over 0.5 Pa s$^{-1}$ and a
southwesterly airflow prevails at high speeds below 400 hPa. The airflow is in the opposite direction at
low speeds above 400 hPa. The maximum ascending speed in the eye wall reaches −1.5 Pa s$^{-1}$ (Fig. 6e).
Figure 6f shows the transportation of water vapor in the typhoon system along line A–B. There is strong
convergence of water vapor below 700 hPa and the maximum WVFD is −5×10$^{-7}$g hPa$^{-1}$ s$^{-1}$ cm$^{-2}$. There
exists a belt of divergence in the middle of the convergence field with WVFD values ranging from 2×10$^{-7}$
to 5×10$^{-7}$ g hPa$^{-1}$ s$^{-1}$ cm$^{-2}$. Water vapor is usually transported in the lower layers of the atmosphere.
Water vapor is exchanged between the typhoon and the eye wall, so the intensity of precipitation is
different at various locations in the typhoon system.
**3.4.2 Frontal precipitation in eastern China**
In frontal systems, precipitation is induced by the elevation of air masses when two types of air flow
meet. As an example, frontal precipitation case that occurred in eastern China (30°–40° N, 112°–122° E)
on 22 June 2003 at orbit 31926 is analyzed.

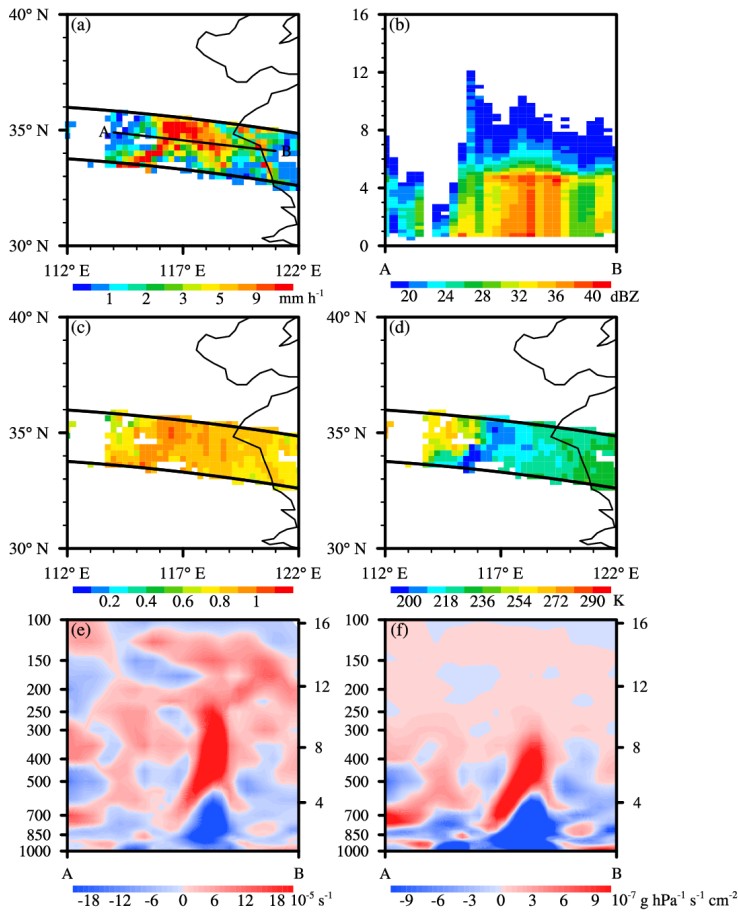

**Figure 7: Case study of frontal precipitation on 3 June 2003 (orbit number 31926). (a) The near-surface rain rate, vertical cross-sections of (b) the precipitation reflectivity factor, (c) RF1, (d) $TB_{10.8}$, vertical cross-sections of (e) divergence and (f) the divergence of water vapor flux along line A-B in panel (a).**

Fig. 7a clearly shows the rain belt at the boundary of the different airflows. The maximum precipitation intensity is more than 10 mm h$^{-1}$. Fig. 7b shows the precipitation reflectivity factor profiles of line A–B to illustrate the vertical structure. The reflectivity factor is high from the land surface to 5 km and the maximum exceeds 42 dBZ. The size of precipitating cloud droplets in the heavy rainfall system is usually small near the precipitating cloud-top, as shown by RF1 values bigger than 0.8 (Fig. 7c). $TB_{10.8}$ varies between 200 and 235 K in the area of strong precipitation intensity and the minimum appears in the front, where the height of the precipitating cloud-top is highest. The difference in the brightness temperature on each side of the front represents the difference in the height of the precipitating cloud-top (Fig. 7d).

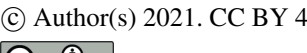



Figure 7e shows that the calculated divergence in the cross-section, which represents convergence below
700 hPa, reaches $-18 \times 10^{-5}$ s$^{-1}$. Strong divergence over $18 \times 10^{-5}$ s$^{-1}$ occurs above 500 hPa. The WVFD
has the same distribution as the divergence. The maximum WVFD exceeds $-10 \times 10^{-7}$ g hPa$^{-1}$ s$^{-1}$ cm$^{-2}$
below 700 hPa. Strong divergence of the water vapor occurs above 500 hPa, with a WVDF of $9 \times 10^{-7}$ g
hPa$^{-1}$ s$^{-1}$ cm$^{-2}$. The exchange of water vapor often occurs below 200 hPa (Fig. 6f). The atmospheric
status satisfies the precipitation condition and shows the complexity of vertical motion in the atmosphere.
**4  Discussion**
Due to the rapid development of the technology of satellite and emergence of the various satellite data
products. Huge amount of the satellite datasets usually leads to difficulty in data storage. The M-1B01-
2A25-GD is a grid-level dataset with spatial resolution of 0.25° which can largely reduce the digital
storage space required. This new dataset will become a demonstration for satellite data processing.
The data merging among the precipitation parameters (profiles of rain rate and precipitation reflectivity
factor, near-surface rain rate and rain type) measured by the PR, spectral signals measured by the VIRS
and atmospheric parameters (temperature, specific humidity, wind field, geopotential height, divergence
and vertical velocity) of the ERA5 reanalysis dataset is an initial attempt. This dataset can be helpful in
studying the characteristics and changes in precipitation and the clouds systems. To further explore the
relationship among the precipitation, clouds and atmospheric parameters, the cloud parameters based on
the signals retrival from the TRMM VIRS data will be added to the merged dataset. The work is now in
progress and will not be involved in this study due to the limited length of the paper.
The studies on the atmospheric dynamics and cloud physics are isolated because of the lack of the suitable
datasets. Now, the problem can be solved through establishing the new merged dataset. The
comprehensive parameters about precipitation, clouds and atmosphere can be obtained in each single
orbit from the M-1B01-2A25-GD to support the case analysis and model simulations.
**5  Data availability**
The used M-1B01-2A25-GD in this paper are accessible at http://doi.org/10.5281/zenodo.4458868 (Sun
and Fu, 2021).



## 6 Conclusions

We establish a new merged gridded dataset M-1B01-2A25-GD combining satellite and reanalysis datasets. The precipitation, cloud and atmospheric parameters are spatially and temporally collocated. The gridded data helps to reduce the digital storage space required. The statistical results show that there is no obvious bias in the 1B01-2A25-PMD when compared with the original swath-level data measured by the TRMM VIRS. The 1B01-2A25-GD has the same spatial resolution as the ERA5 reanalysis dataset. The average inside the grid leads to smoothing effects on the maximum and minimum values, but does not adversely influence the parameters in the 1B01-2A25-PMD. The difference in the mean value is no more than 0.87 and the STD is no more than 2.38 for the near-surface rain rate and signals measured by the VIRS over one orbit. The M-1B01-2A25-GD contains comprehensive parameters about precipitation, clouds and the atmosphere that are useful in studies of the characteristics and distribution of precipitation and clouds systems in the tropics and subtropics. Three typical applications of the M-1B01-2A25-GD are introduced by analyzing different examples of precipitation. This new dataset can support studies of precipitation, clouds systems and model simulations. Longer time periods of data and more parameters will be added as satellite technology and models are improved.

**Author contribution.** Lilu Sun and Yunfei Fu prepared the data in the standardized format. Lilu Sun uploaded the data in the data repository and prepared the manuscript with contributions from all co-authors.

**Competing interests.** The authors declare that they have no conflict of interests.

**Financial support.** This research was supported by the National Natural Science Foundation of China (Grants 91837310, 41675041).

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
