# Peer review of "A new merged dataset for analyzing clouds, precipitation and atmospheric parameters based on ERA5 reanalysis data and the measurements of TRMM PR and VIRS"

_Earth System Science Data, 2021_

## Author Response (AR1)

**Responses to RC**

We are grateful to the Editor and the Reviewers for reviewing our manuscript. The comments and suggestions are very helpful and valuable. Kindly find a point-by-point reply to the issues as follows.

**RC1:**

1. Line 167, "weight-average method":

1) How to calculate weight in the "weight-average method"?

2) When the spatial resolution of the data is 5km, the signal is the average value of 5 km. When the spatial resolution is 2.5 km, the signal is the average value of 2.5 km, so it is better to use the simple average method or the weighted average method?

Response: Thank you for your questions.

(1) Due to the different spatial resolutions of the TRMM VIRS (2.2 km) and PR (4.5 km) pixels. There are usually about seven VIRS pixels near one 2A25 pixel in the same orbit. "Weight" here means "inverse distance weight". For each VIRS pixel within one PR pixel area, according to the distance between the VIRS and PR pixels, the spectral signals of the TRMM VIRS are calculated by weighted averaged in the 1B01-2A25-PMD dataset. (Fu et al., 2011)

(2) It is better to use the weighted average method. The spatial resolutions of the TRMM VIRS and PR pixels are different. Also, the relative location between the TRMM VIRS and PR pixels changes slowly due to the scanning angle and direction of the satellite. In order to obtain merged data, the resolution of the TRMM VIRS pixel should be reduced to 4.5 km which is the same as the PR pixel. Compare to the simple average method, the latter method is more reasonable in the data merging process theoretically.

2. Line 173:

It should be described that the spatial resolution of 1B01-2A25-PMD is 5 km and the data width is 220 km

Response: Thank you for your advice. We have added the sentences in the revised manuscript. [Line 177-178]

**3. Line 195:**

Atmospheric status or parameters

Response: Thank you. We have changed "the atmosphere" to "the atmospheric status". [Line 200]

**4. Lines 214-215:**

The difference in TB10.8 is also big.

Response: Thank you for pointing out the phenomenon.

The difference in  $TB_{10.8}$  and  $TB_{3.7}$  signals may be caused by the uneven cloud distribution. The difference in  $TB_{3.7}$  is more obvious than that in  $TB_{10.8}$ , so we made further explanations on the difference in  $TB_{3.7}$ . The results shown in the Fig 2 can explain the mentioned phenomenon. We have added the descriptions on the difference in  $TB_{10.8}$  in the revised manuscript. [Line 220-222] "Also the difference in  $TB_{10.8}$  signal is noticeable. Although the ranges and the maximum of the  $TB_{10.8}$  nearly unchanged after merging, but the shape of the PDF line is different near the maximum."

5. Lines 251-256:

1) Figure 3 has proved that the difference after gridding is small, and there is no need to count the difference between the data.

2) The previous analysis is a special case (using a certain track data) analysis, and the statistical analysis is preferably an overall analysis, such as counting the average value of data differences over several years.

Response: Thank you for the suggestions.

(1) We present the table about the comparison of the parameters between the 1B01-2A25-PMD and 1B01-2A25-GD to quantify the influence of the gridding process.

(2) In this work, we focus on the characteristics of the single orbit data. Due to the limited length of the paper, the treatment on a longer period of data will not be included. The overall analysis of the data differences over several years will become the main point of our next step work.

6. Line 262:

The M-1B01-2A25-GD data is not evaluated in this manuscript? How to evaluate this data?

Response: Thank you. According to the previous evaluations of the 1B01-2A25-PMD and 1B01-2A25-GD, the parameters of precipitation, cloud and atmospheric status are reliable in the finally merged dataset. The practical use and presentation of the parameters are mainly included in this section. Within the same orbit field, both precipitation, cloud and atmospheric parameters can be presented at the same time. We have changed the title of this section to "Presentation of the M-1B01-2A25-GD parameters". [Line 270]

**7. Line 410:**

There is only one track of data, which cannot explain the overall error situation.

Response: Thanks. We have changed the description in the revised manuscript to avoid the misunderstanding. [Line 415] "In the arbitrarily chosen orbit, the difference in the mean value is no more than 0.87 and the STD is no more than 2.38 for the near-surface rain rate and signals measured by the VIRS."

All the above have been modified in the revised manuscript. Thank you again.

Fu, Y. F., Liu, P., Liu, Q., Ma, M., Sun, L., and Wang, Y.: Climatological Characteristics of VIRS Channels for Precipitating Cloud in Summer Over the Tropics and Subtropics, Journal of Atmospheric and Environmental Optics (in Chinese), 6, 129-140, https://doi.org/10.3969/j.issn.1673-6141.2011.02.009, 2011.

**RC2:**

1. Introduction, I suggested the authors to pay more attentions to the development of merging method for the multiple data sources, the available merging data sets, and the corresponding results in accuracy evaluation.

**Response: Thank you for your advice.**

In the introduction part, we gave three main aspects. First, the importance of the 3D structure of the precipitation and clouds are showed which can explain why we established this new dataset. Then, introducing the datasets which can be used to study the characteristic of the precipitation and clouds system. So that we can understand which datasets can be chosen to realize the aim. Finally, combining dataset of different types becomes a developed tendency in the study and we combine those datasets from the TRMM PR, VIRS and ERA5 reanalysis.

Those three aspects are equally important. To enrich the content in the introduction part. We have added some related references in the revised manuscript. [Line 89-93] "Fu et al. (2013) used moving surface fitting method to combine TRMM TMI and PR pixel data. The differences in the mean, standard deviation and frequency distribution between the original and merged data are analyzed to validate the accuracy. Wang et al. (2017) merged the TRMM PR 2A25 products with the IGRA dataset to investigate the profiles of temperature and humidity for the convective and stratiform precipitation."

Generally, as you mentioned that there are many data merging methods, but this paper focuses on merging satellite data with reanalysis data to obtain the corresponding atmospheric parameters of the precipitation structure detected by satellite-borne precipitation radar, so as to provide merging data for the subsequent research on the atmospheric environment of the precipitation structure. Thanks!

2. Data, to make it easier to follow the involved data set, it is preferred to list the critical informations (e.g. spatial resolution, span period, recorded frequency) using a table.

Response: Thank you for your nice suggestion. We have added the corresponding table in the revised manuscript. [Line 201]

| M-1B01-2A25-GD      |                                                                                                                                                                                                                                                      |
|---------------------|------------------------------------------------------------------------------------------------------------------------------------------------------------------------------------------------------------------------------------------------------|
| Spatial resolution  | 0.25°                                                                                                                                                                                                                                                |
| Temporal resolution | Hourly                                                                                                                                                                                                                                               |
| Main parameters     | profiles of rain rate and precipitation reflectivity
factor, near-surface rain rate, rain type, spectral
signals measured by the VIRS, temperature,
specific humidity, wind field, geopotential
height, divergence and vertical velocity |
| Swath width         | 220 km                                                                                                                                                                                                                                               |
| Vertical coverage   | 0 to 20 km (precipitation profile)                                                                                                                                                                                                                   |
|                     | 1000 hPa to 10 hPa (atmospheric parameters)                                                                                                                                                                                                          |
| Vertical resolution | 0.25 km (precipitation profile)                                                                                                                                                                                                                      |
|                     | 32 pressure layers (atmospheric parameters)                                                                                                                                                                                                          |

**Table 1. Critical information of the M-1B01-2A25-GD**

3. Method, the current description is insufficient. How to merge the datasets? How to deal with the so-called "match", how to evaluate the reality of merged product.

Response: Thank you for your nice questions.

(1) There are three main steps in the data merging process. First, we merged the 1B01 and 2A25 pixel data due to the little time lag between the TRMM VIRS and PR. We reduced the spatial resolution of the 1B01 pixels to merge with the 2A25 pixels at the same orbit. Second, to match with the ERA5 reanalysis data, the merged pixel data should be gridded, the resolution of the gridded data is 0.25 degree. Third, ERA5 data has the hourly output, so for the same grid location and time, we merged the 1B01-2A25-GD with the ERA5 reanalysis data.

(2) "Match" here in this paper refers to the process of the data merging in a spatial and temporal way.

(3) As can be seen in section 3.1, after merging process between the 1B01 and 2A25 pixel data. Fig 1 reveals that the PDFs of RF1, RF2 and  $TB_{12.0}$  are almost the same before and after data merging. The differences on the PDFs of  $TB_{3.7}$  and  $TB_{10.8}$  can be explained from the results shown in Fig 2. Also, Fig 3 shows that no unacceptable distort exists on the 1B01-2A25 merged data after gridding process. Three different types of the precipitation and clouds cases were presented in section 3.4. Three-dimension of the precipitation, clouds and atmospheric parameters were plotted to fully understand the characteristics of the cases.

4. What's the value for the developed dataset, what gaps can be filled comparing to the available dataset?

Response: Thank you for your nice questions.

First of all, the developed dataset contains comprehensive parameters, such as profiles of rain rate, precipitation reflectivity factor, spectral signals and atmospheric parameters. Also, the grid dataset has fine spatial resolution about 0.25 degree and hourly temporal resolution, which filled a gap in this field.

Compare to the available datasets, the dataset introduced in this paper can be very useful for analyzing the characteristics of the precipitation structure and its atmospheric environment in precipitation system. The dataset provides us the comprehensive parameters simultaneously among the same orbit field with less digital storage space.

All the above have been modified in the revised manuscript. Thank you again.